# Biomolecules from Macroalgae—Nutritional Profile and Bioactives for Novel Food Product Development

**DOI:** 10.3390/biom13020386

**Published:** 2023-02-17

**Authors:** Laura E. Healy, Xianglu Zhu, Milica Pojić, Carl Sullivan, Uma Tiwari, James Curtin, Brijesh K. Tiwari

**Affiliations:** 1Teagasc Food Research Centre, Ashtown, D15 DY05 Dublin, Ireland; 2School of Food Science and Environmental Health, Technological University Dublin, D07 EWV4 Dublin, Ireland; 3School of Biosystems and Food Engineering, University College Dublin, National University of Ireland, Belfield, D02 V583 Dublin, Ireland; 4Institute of Food Technology, University of Novi Sad, Bulevar cara Lazara 1, 21000 Novi Sad, Serbia; 5Faculty of Computing, Digital and Data, School of Mathematics and Statistics, Technological University Dublin, D07 EWV4 Dublin, Ireland; 6Faculty of Engineering & Built Environment, Technological University Dublin, D07 EWV4 Dublin, Ireland

**Keywords:** new product development, seaweed aquaculture, seaweed processing, sustainable protein

## Abstract

Seaweed is in the spotlight as a promising source of nutrition for humans as the search for sustainable food production systems continues. Seaweed has a well-documented rich nutritional profile containing compounds such as polyphenols, carotenoids and polysaccharides as well as proteins, fatty acids and minerals. Seaweed processing for the extraction of functional ingredients such as alginate, agar, and carrageenan is well-established. Novel pretreatments such as ultrasound assisted extraction or high-pressure processing can be incorporated to more efficiently extract these targeted ingredients. The scope of products that can be created using seaweed are wide ranging: from bread and noodles to yoghurt and milk and even as an ingredient to enhance the nutritional profile and stability of meat products. There are opportunities for food producers in this area to develop novel food products using seaweed. This review paper discusses the unique properties of seaweed as a food, the processes involved in seaweed aquaculture, and the products that can be developed from this marine biomass. Challenges facing the industry such as consumer hesitation around seaweed products, the safety of seaweed, and processing hurdles will also be discussed.

## 1. Introduction

### 1.1. What Is Seaweed?

Seaweed, or, macroalgae, are a plant group of multicellular algae that have been exploited on a large-scale for the extraction of valuable functional food ingredients (primarily alginates, agar, and carrageenan) [1]. However, interest in the whole crop as a source of nutrition for humans is gaining more interest due to its nutritional properties [2]. The potential of seaweed is two-pronged in its cultivation and market opportunities. For instance, seaweed farming, known as seaweed aquaculture, can occur in any suitable coastal region, with relatively cheap input costs to set up. The marketplace, at the end of the seaweed aquaculture production line, is seeing huge growth in demand for sustainable, plant-based products [3].

### 1.2. Nutritional Importance of Seaweed

As a food, seaweed is included in about 21% of meals consumed in Japan [4] while, in South Korea, the intake of seaweed was recorded as approximately 255 g/person/month in 2009 [5]. Algal polysaccharides such as fucoidan, alginate, laminarin, agar, carrageenan, and ulvan are important features of the seaweed nutritional and bioactive profile [6], not just for human health, but as added techno-functional food ingredients and in sustainable packaging solutions [7] for the human food chain. Understanding the different nutritional qualities of various species of edible seaweed will be important for getting products to market. Protein content can vary widely within the three seaweed groups such as in green seaweed (90–184 g kg^−1^ dw), red seaweed (103–201 g kg^−1^ dw), and brown seaweed (59–120 g kg^−1^ dw) [8]. Table 1 highlights protein variation in some seaweed species.

Carbohydrate content (usually the dominant component of every seaweed species), mineral content and amino acid profiles are variable across species and groups of seaweed [8]. Table 2 shows the wide variations in amino acid content that can be observed within the group of brown seaweed itself, with glutamic acid ranging from 18.4–55.34 mg g^−1^ dw across three species.

Carbohydrate content also varies across species [10] and these sugars (such as mannitol) are valuable in multiple applications from functional food ingredients to biorefinery processing. These properties are important to understand for the processing of seaweeds, and indeed in the development of the products that can be made from them. Evaluating the best use of each seaweed group is not reliable due to the wide-ranging variation among species across all groups of seaweed. However, these variations are essential to consider when determining the best route to market with a particular seaweed product.

### 1.3. Relevance of This Review

This review paper is divided into three sections: (i) seaweed properties (why is seaweed consumed), (ii) seaweed processes (how is seaweed transformed from plant to plate), and (iii) seaweed products (how is seaweed consumed). Section one includes the nutritional profile of seaweed with a focus on iodine and its impact on the final products. The second section includes different methods of processing with novel applications in the context of seaweed. Lastly, section three discusses a range of seaweed products on the market and in development. As a narrative review, this paper has compiled a comprehensive, critical, and objective analysis of the topic of seaweed aquaculture, through an extensive review of the available literature. The aim of this narrative review is to provide relevant and timely information on the above areas of novel food product development for food producers facing pressure in this area. The topics covered in this review will provide the reader with a full picture of the state-of-the-art of the seaweed aquaculture industry. Figure 1, below shows an overview of seaweed aquaculture and some of the topics discussed in this review paper.

## 2. Seaweed Properties

Seaweed is an excellent source of protein, minerals, vitamins, soluble, and insoluble fibre, as well as a range of bioactive molecules. There is a range of potential benefits to be derived from the consumption of seaweed as part of our normal diet. Seaweed has unique nutritional offerings for the human food market.

### 2.1. Seaweed Nutritional Properties

#### 2.1.1. Protein

As the global population expands, we are observing an increase in demand for protein. It is expected that novel protein sources will become more common in the European food and feed markets and will likely include insects, algae, and duckweed [11]. Other sources of protein that are currently consumed by humans are expected to expand their utility as our demand for plant-based protein increases.

Many of the proposed plant-based “alternative protein” sources are already on the market. For example, Quinoa is a pseudocereal that is lower in carbohydrates and higher in protein content (12–23% protein) than major cereals like wheat and maize [12]. Hemp is another plant source that has many applications and can be considered a valuable alternative protein source for the human food market due to its quality protein profile [13]. Other potential sources such as legumes and lentils are also growing in demand [14,15]. The revalorization of byproducts such as rice bran spent brewer’s grain, oil seed cakes and pulse byproducts will greatly expand the availability of alternative protein sources to exploit [13].

Seaweeds are a promising source of protein (Table 1) [16], with a variable protein content that ranges from 5–47% across species [17]. Green and red seaweeds contain more protein (10–47%) than brown (5–24% DW) [16,18]. In a study examining the nutritional composition of red seaweed *Grateloupia turuturu*, seasonal variation of protein content was found, having implications for producers who wish to state standardized nutritional claims on their packaging [18]. Within seaweed species, protein can vary seasonally such as in red seaweed *Palmaria palmata* where protein content can vary from 8–35% [19]. Seaweed is a valuable source of amino acids (Table 2), particularly glycine, alanine, arginine, and proline [17], as well as aspartic and glutamic acids [20].

Of fourteen seaweed species investigated by Ramos et al. (2000), six species *(Amansia multifida, Bryothamnion seaforthii, Bryothamnion triquetrum, Corallina ofticinalis, Digenea simplex*, and *Enantiocladia duperreyi*) were reported to have high levels of lysine. Lysine is an essential amino acid that plays an important role in a number of biological processes such as development and cellular differentiation, as well as disease [21]. As trends towards more plant-based diets continue, it is important that lysine requirements are met by sources other than meat. Consumption of seaweed could compensate as potential ingredients for incorporation into lysine-deficient cereals [22]. L-lysine can be extracted from seaweeds such as *L. digitata* using the principles of biorefinery, where the residual sugars (namely, mannitol) following a commercial alginate extraction method are converted to L-lysine using metabolically engineered *Corynebacterium glutamicum* [23].

However, some seaweed species also reported for their limiting amino acids. For instance, tryptophan was the first limiting amino acid found for all species studied by Dawczynski et al. (2007), with leucine and isoleucine found to be limited in red species while methionine, cystine, and lysine were found to be limited in brown species [24]. The same study reported that aspartic and glutamic acid are the most concentrated amino acids across a range of 34 products that contained five different seaweed species [24]. Looking closer, brown algae was reported to contain significantly higher amounts of these amino acids compared to red algae [24]. The aspartic acid content of the *Laminaria* species (brown seaweed) was reported to contain a high amount (12.5 ± 2.8 g/16 g N) compared to other brown species tested, namely *Undaria pinnatifida* (8.7 ± 1.1 g/16 g N) and *Hizikia fusiforme* (9.l ± 1.0 g/16 g N) [24]. Table 2 shows the variations in amino acid content observed within brown seaweeds. Compared to brown species, red seaweed species have similar essential amino acid (EAA) concentrations [24].

Moreover, Mæhre et al. (2014) also reported that red, green and brown seaweed species contained all EAA (essential amino acids) in sufficient amounts, except for methionine and semiessential cysteine [25]. The study highlights that a limiting amino acid like methionine was still found to be three times higher in seaweed species compared to soy [25].

For fish feed, supplementation of fish meal with seaweed to combat the limiting of amino acids could enhance the nutritional value of the product such as the development of fish feed for fish farming [25]. With this in mind, seaweed offers an opportunity to food innovators to develop healthy plant-based protein products (refer to Section 4 for further information).

#### 2.1.2. Fatty Acids

*Alaria esculenta* is a well-documented brown seaweed species high in amino acids as well as PUFAs (polyunsaturated fatty acids) like eicosapentaenoic acid (EPA) and stearidonic acid [26,27]. The omega-6:omega-3 ratio is an important metric for measuring benefits to health and this value should be below 10 in order to provide anti-inflammatory benefits and prevention of cardiovascular and nervous system disorders associated with these fatty acids [28]. For instance, a ratio value of 1.93 (*Alaria esculenta)* and 1.00 (*Saccharina latissima)* was reported by [27].

#### 2.1.3. Polysaccharides

Making up to 60% of the nutritional profile of seaweeds, carbohydrates are a prominent feature in seaweeds that can drive the end application of the species used [29]. Seaweed polysaccharides have long been extracted for the techno-functional ingredients market, as discussed in further detail in Section 2.3.

Sulfated polysaccharides are found in seaweed and have been shown to have anti-inflammatory and antioxidant activity, including the ability to scavenge free radicals [30]. The applications of such valuable nutraceuticals are being put forward to treat a number of diseases, including Alzheimer’s disease [31]. Scavenging activity was found to increase with an increase in the polysaccharide concentration of seaweed [10].

Bauer et al. (2021), observed that the brown seaweed *Halidrys siliquosa* has a very high proportion of a polysaccharide, mannitol (200 g kg^−1^ dw), that can be exploited for fermentation [31] in biorefinery settings. Carbohydrate content is an important metric for the biorefinery industry and should be comparable with lignocellulosic biomass in order to be a sufficient biomass for the process [31]. Due to their high carbohydrate content, *Saccharina latissima* and *Laminaria digitata* are deemed suitable candidates for biorefining [31].

#### 2.1.4. Fibre

Fibre is an important nutrient for the human body and is required for digestion and sustaining a healthy gut microbiome [32]. Sources of dietary fibre come from plant polysaccharides that are “resistant to digestion and absorption in the small intestine but undergo complete (or partial) fermentation in the large intestine” [33]. Due to relatively recent advances in our understanding of the importance of our gut microbiota and its impact on our overall wellbeing, there is increasing demand for foods which promote a healthy gut population.

Seaweed is rich in dietary fibre [34]. Dietary fibre content varies greatly between seaweed species (25–75% of dry weight) and is primarily soluble [35]. Patarra et al. (2011) studied a range of eight brown seaweed species for fibre content and found that fibre content ranged from 33.82 to 63.88% [36]. The same study also analysed protein content and found that species with lower protein, such as *Fucus spiralis* (10.77% protein) had higher fibre contents (63.88% in *Fucus spiralis*) [36]. Laminarin, a beta-polymer of glucose, is considered a source of dietary fibre found in seaweeds as it is undigested in the upper digestive tract, and will be discussed in Section 2.2 [33].

#### 2.1.5. Minerals and Vitamins

A high-quality mineral profile is an important feature of any food product. Postharvest treatment, such as blanching and drying, can impact upon the mineral profile of seaweed [37]. Zinc is a crucial mineral for human health, and is involved in maintaining a healthy immune system among other roles [38]. Seaweed is a source of zinc for humans, with red seaweeds reportedly having the highest quantity of the mineral when compared with brown and green seaweeds [39]. However, the red seaweeds also had the widest range of zinc, containing both the highest (*Mastocarpus stellatus*) and lowest (*Corallina officinalis*) values [39]. In addition to zinc, seaweeds like *Ulvan fascitya* and *Gracilara edulis* are also important sources of iron (14.8–72 mg/100 g), iodine (38.8–72.2 mg/100 g), and calcium (410–870 mg/100 g) [40].

Cabrita et al. (2016) examined different seaweed species (red and brown species) as an animal feed and found seaweed to be a valuable source of minerals like calcium, magnesium, iron, iodine, copper, manganese, and selenium and low in phosphorous and zinc [41]. Like other nutritional quality aspects of seaweeds, it is important to standardize species, environmental conditions, and season to ensure consistency in the quality of the end-product.

Seaweeds are also a good source of vitamins such as B_1_, B_2_, B_12_, C, beta-carotene, and E [42]. These vitamins are known to be crucial to human health with immune support (vitamin C), cardiovascular disease protection (beta-carotene), and cancer prevention (vitamins E and C), among many other human-health functions [42]. In this way, seaweed can be used as an ingredient to enrich common food products as a means of enhancing the nutritional profile and delivering value-added health benefits to the consumer (refer to Section 4). A comprehensive review of vitamin C content across 92 species of seaweed was carried out by Nielsen et al. (2014), and reported that the average vitamin C content was 0.773 mg g^−1^ dw (dry weight) with the highest content found in the red seaweed species *Hydropuntia edulis* (>3.00 mg g^−1^ dw) [43]. They reported no significant difference in vitamin C among the three main taxonomic groups of seaweed [43].

Seaweeds are an especially good source of iodine [44]. Iodine is an essential mineral nutrient, crucial for normal thyroid function when consumed in the adequate dosage. The daily intake requirement of iodine is about 2 μg kg^−1^ of body weight, i.e., about 140 μg for a (70 kg) adult [45]. It has been long understood that a diet deficient in iodine can cause adverse hormonal and hypothyroidal effects, leading to iodine being recommended as an essential daily mineral as part of a healthy, balanced diet. Contrastingly, too much iodine intake can lead to hyperthyroidism, and therefore there is some concern relating to seaweed inclusion in the human diet [45].

Seaweed can be a valuable source of iodine for human and animal diets, with the intake of just one sushi meal containing seaweed being associated with increased iodine excretion, bringing the level of iodine in participants with a mild iodine deficiency to sufficient levels [46]. There are some concerns relating to iodine accumulation in seaweed leading to potentially harmful levels of iodine in the end products (refer to Section 2.5.1).

### 2.2. Seaweed Bioactives Profile—Is Seaweed a Superfood?

There is increasing interest from Western civilizations in low-dose, high-concentrate health foods, often interlinked using ingredients from “traditional”, sometimes exotic, sources. Due to their “functionality”, these foods are often accompanied by a premium price, although the active ingredients are usually minerals, antioxidants, or n-3 fatty acids [47]. Seaweeds are one such source of these bioactives (such as natural antioxidants, minerals, and high-quality PUFAs) making them prime candidates for the superfood market [47]. Direct health-benefits of utilising seaweeds as a treatment have been determined by a number of studies. Fucoidan, a sulfated polysaccharide found in seaweed, has known anticancer mechanisms such as cell cycle arrest, induction of apaoptosis, antiangiogenesis, and anti-inflammatory [48].

Seaweeds are an important source of polyphenols such as phlorotannins, bromophenols, flavonoids, and phenolic terpenoids as highlighted by [49]. These compounds have known benefits such as antitumor, antidiabetic, antifungal, antimicrobial, antithrombotic, antiviral, immunomodulatory, neuroprotective and anti-inflammatory activities [49].

Phlorotannins are a group of phenolic compounds that are synthesized by brown algae [50]. The extraction of such compounds to make value-added products can provide healthy, superfood products. However, in the case of phlorotannins, there may be limits to their applications due to the associated bitterness profile of tannins [51]. Novel food developers need to be innovative in delivering a good-tasting product without damaging the molecule and ensuring its arrival to the designated part of the digestive tract for absorption [51]. Green technologies such as supercritical fluid extraction and enzyme-assisted extraction are suitable ways of obtaining phlorotannins from seaweed [51]. Limitations in the mass application of phlorotannins as a functional food ingredient relate to scaling issues, affordable technology, and a lack of “comprehensive knowledge of its chemical characterization” [51].

Laminarin is one such molecule under investigation as a health supplement for humans. It is a polysaccharide and energy-store located in the vacuoles of brown seaweed cells [52]. Different species contain different amounts of laminarin and are also affected by the season of harvest and extraction methods [52]. Research has shown its promising applications in cancer therapy, tissue engineering, as an anti-inflammatory, and as an antioxidant, as described in detail by Zargarzadeh et al. (2020) [53].

Fucoxanthin is a carotenoid found as an accessory pigment in the chloroplasts of brown algae with major potential health benefits [54]. Research is being carried out into the exact mechanism of the bioactive, however, a number of biological functions have already been established such as anticancer, antioxidant, antihypertensive, anti-inflammatory, antiobesity, and radio-protective activities [55].

Fucoidan is a sulphated polysaccharide which can be found mostly in the cell-wall matrix of brown seaweed species [56]. A lot of studies now back up the benefits of fucoidans and fucose-rich sulphated polysaccharides, which include immunomodulation, anti-inflammatory, anticancer agent, and antiviral properties, among other benefits [48]. In a study determining the scavenging activity of polysaccharides extracted from seaweeds, the brown seaweed species *Padina gymnospora* was found to have the highest percentage, followed by the red seaweed species *Kappaphycus alvarezii*, and lastly the green seaweed species *Kappaphycus striatus* [10].

### 2.3. Seaweed as a Techno-Functional Food Ingredient

Seaweed is used as an ingredient in many household products such as cosmetics [57], foods like ice cream [58], and as a stabiliser in beverages like wine [59]. Seaweed is not only a stand-alone whole food or food ingredient, it is also a complimentary additive to other foods with the potential to offer important functional properties. Hydrocolloids are a group of molecules with such functional properties. Seaweeds are a valuable source of hydrocolloids, namely alginate from brown seaweeds and carrageenans and agar from red seaweeds [60]. Hydrocolloids are a heterogeneous group of long-chain polymers characterized by their ability to form viscous dispersions and/or gels when in water [61]. As seen in earlier sections, the seaweed nutritional profile widely varies and the same can be said for polysaccharide content, where it can be as high as 76%, however, it usually averages 50% [62]. Hydrocolloids are naturally occurring compounds found in a range of sources (Table 3).

In the baking industry, hydrocolloids are a much sought-after ingredient as they can enhance the end product quality significantly as a bread improver, in dough handling, prolonging the freshness of the bread, and extending shelf life [64]. Alginate is found in the cell walls of brown algae in the form of calcium, magnesium, and sodium salts of alginic acid [65]. This biomolecule has unique properties that could offer enhanced flour applications. Agar is also used in the baking industry for its gelling properties and high temperature resistance [61]. It is a food-grade polysaccharide extracted from red algae for its gelling and stabilizing properties. *Gelidium* and *Gracilaria* genera are the most commonly extracted seaweeds for obtaining agar [65]. Agar can form a gel at concentrations as low as 0.5–2% over a wide range of pH [62]. Carrageenan can be extracted from a group of red algae in the class *Rhodophynceae* [62]. Concentrations of carrageenan in algae are affected by season, and growth conditions, and can even vary in structure between species [62]. Food applications of carrageenan include gelling, thickening and stabilizing, and are used in two main categories. water based and dairy products [62].

As discussed later in Section 4, extracted seaweed molecules are being investigated for use in biodegradable packaging. Due to the nonbiodegradable nature of plastics, and their subsequent environmental damage, earth-friendly alternatives are much sought after. Ramadhani et al. (2019) produced an edible, biodegradable film from seaweed powder (*Eucheuma cottonii* sp.) that was most effective when included with the plasticizer glycerol (at 0.5%) [7]. A quality edible film, such as those made from hydrocolloids, should have the ability to protect internal products against exposure to oxygen and carbon dioxide while keeping the product intact and ensuring structural integrity [7]. Edible films such as those made from hydrocolloids extracted from seaweeds are desirable for their environmental benefits and also for human consumption as they offer a nontoxic, safe way to consume ready-to-eat products.

### 2.4. Sensory Properties and Consumer Perception

Glutamic acid is an amino acid found in many seaweeds, with the highest concentrations found in brown seaweeds [66]. It is known to be responsible for the “umami” taste in seaweed, and because of this, seaweeds can be used as a source of obtaining this flavour for foods [66]. This “free glutamate” was first extracted in Japan from a dish known as dashi, a seaweed broth [67]. Umami is now an important “fifth basic taste” and is sought after as a means of enhancing all types of dishes, with a particular interest in vegan and vegetarian food products [67].

Some consumers of seaweed may be apprehensive about a “fishy” or “seafood” taste. Although many sensory tests don’t report this issue, Du et al. (2021) found that the removal of the compounds that cause this flavour have been successful [68]. In a study by Losada-Lopez et al. (2021) examining consumer’s perception of seaweed, the consumer’s willingness to consume seaweed was negatively affected by neophobia [69]. However, they noted that the promotion of the health and natural attributes of seaweed as a food source could be successfully used to engage consumers, particularly those interested in new experiences [69]. Another study found that seaweed was most popular when consumed as a snack, or as an ingredient in bread or other dishes [70]. Sustainability was cited as a main reason for participants in this study to buy seaweed-based products [70].

Simone (2020) conducted a study evaluating seaweed consumption across demographic variables and found that the trend of consuming seaweed was associated with younger age, higher income, and higher education levels [71]. When incorporated into food products like biscuits, chips, and juices, seaweed was found to be more acceptable to consumers than seaweed alone [72]. Furthermore, Al-Thawadi (2018) reported that sensory appeal, perceived healthiness, and knowledge or experience with seaweed have a direct effect on the consumer’s behaviour [72]. Therefore, these aspects of marketing should be optimized. The addition of two seaweeds, *Ascophylum nodosum* and *Chondus crispus* to whole-wheat bread was analysed for consumer preference at different inclusion rates; 2%, 4%, 6% and 8% [73]. The bread containing *A. nodosum* was most preferred when it was included at a rate of 4%, while *C. crispus* was most preferred at 2%. However, in both cases, the control bread was the most preferred. The authors also found that as the percentage of seaweed increased, unfavourable features such as saltiness, dryness, denseness and strong aftertaste increased, thus decreasing consumer acceptability of the product [73]. The authors stated that consumers may be willing to eat bread containing seaweed at the low inclusion rates mentioned, and gradually become more likely to consume other products containing seaweed [73,74]. Seaweed acceptance by consumers suffers from a major cultural divide, where some countries or regions possess strong traditions of seaweed consumption and other areas lacking any history of access or utility of seaweed. Some researchers have stated that consumers have a poor perception of seaweed due to its connection with poverty [11,74].

### 2.5. Safety Aspects of Seaweed

Due to increasing rates of allergies related to some food sources, contamination of food with heavy metals (such as mercury in fish), and persistence of other harmful chemicals in food and their link to human disease, there is pressure on industries such as seaweed aquaculture to produce “clean and green” products. Commonly consumed crops like wheat and soybean are associated with allergies and so a shift away from these monoculture crops is in high demand [75]. Seaweed is not without its disadvantages or health-related queries. In Section 3, some postharvest strategies are discussed that can limit the levels of these potentially toxic elements in seaweed. Due to its nature as a sea-based crop, seaweed is susceptible and exposed to any toxic residues or dangerous organisms that may be present in our water bodies. It is hard to control an environment as large as the sea, and in this way, promoting the cleaning of our waters should go hand-in-hand with the development of new sustainable food systems such as seaweed aquaculture. Seaweed, like any crop, is heavily influenced by its environment and there are concerns relating to its uptake of certain minerals and heavy metals.

Arsenic, cadmium, iodine, and *Salmonella* were identified by Banach. et al. (2020) as the four major hazards affecting the European seaweed chain [76]. They highlight that factors such as seaweed species, season, harvest, and processing environments have an impact on hazard presence [76]. Furthermore, seaweed that is collected or processed in close proximity to other industry activities is identified as a potential pollutant source that can increase the likelihood of these hazards existing in the food chain [76]. *Salmonella* may enter the seaweed food chain during growing, harvesting, or processing as a result of unsanitary conditions at the site or contaminated waterbodies which the seaweed was harvested from [76]. Contrastingly, seaweed is commonly used in animal feed as an antimicrobial agent [77]. The *Salmonella* hazard identified by Banach. et al. (2020) is usually a result of contamination, rather than the susceptibility of seaweed as a carrier of the bacteria [76]. Seaweed has the ability to absorb molecules from its environment, potentially putting consumers at risk. Dioxins, polychlorinated biphenyls (PCBs), and polychlorinated dibenzo-p-dioxins (PCDDs) have been found in seaweeds in industrially polluted areas [76,78].

#### 2.5.1. Iodine

Seaweed can absorb and retain high concentrations of iodine from the environments they live in with some species of brown seaweed known to accumulate over 30,000 times the iodine concentration of seawater [79]. As mentioned in Section 2.1.5, there are issues associated both with too much and too little iodine intake in human diets. It is therefore very important for seaweed producers to clearly state iodine content on their packaging, as well as monitoring and reporting of iodine levels.

#### 2.5.2. Heavy Metals

Arsenic (As) is a known toxin to human life and specifically, inorganic arsenic is monitored due to its known toxicity. Classified as a Group-A carcinogen, long-term human exposure is associated with an elevated health risk, with water sources of this toxin being more dangerous than soil sources to humans due to its increased bioavailability [80]. Rose et al. (2007) examined inorganic arsenic levels across five species of commercial seaweed and found that *Hijiki fusiformis* was the only species to contain inorganic arsenic elevated above the limit of detection [81].

EU Regulation 744/2012 [82] states that As should not exceed 40 mg kg^−1^ (for a moisture content of 12% *w*/*w*, which equates to 45 mg kg^−1^ dry weight) [82]. A study by Afonso et al. (2020) revealed that As content of two species of seaweed*—Alaria esculenta* and *Saccharina latissima*—exceeded this recommendation in a range of 53.11–59.93 mg kg^−1^ dry weight [27]. However, it is important to note that this As content was found to be of organic origin (exceeding 95% of the overall As content) [27]. A study determining the quantification of organic and inorganic arsenic in 20 species of marine organisms including fish, sea urchins, crustaceans, and seaweed found that inorganic arsenic was detected at low levels of 0–7% of the total As content across all species [83]. There was one exception to this low content found in the aforementioned *H. fusiforme*, where inorganic As were found present at 60% of the total arsenic content [83].

The HI (hazardous index) for aluminium content was found to be in excess of the safe limits for children in a review of green seaweeds, according to the US-EPA (United States Environmental Protection Agency) guidelines, suggesting the need for close monitoring of heavy metal content, particularly in green seaweeds [84]. Khandaker et al. (2021) suggest further examination of heavy metals and their accumulation in seaweed is needed, along with the implementation of processes to remove these potential toxins through bioadsorbent techniques [85].

#### 2.5.3. Allergens

In a study by Filippini et al. (2021), the correct labelling of seaweed products is highlighted as an issue for the seaweed market [84]. While most European products were found to be labelled correctly, there were some irregularities in language and allergen declaration. Contaminants such as mollusc shells and small crustaceans were found in some samples which can pose a threat to consumers with allergies. Furthermore, these potential allergens were not declared on the labels assessed by Filippini et al. (2021) [84]. Seaweed products from Europe are obligated to follow labelling legislation, however, this paper highlights that caution needs to be taken when consuming seaweed products of a non-European origin due to a lack of mandatory labelling of allergens [84].

#### 2.5.4. Salt Content

A diet that is high in salt is associated with a significantly increased risk of stroke and cardiovascular disease [86]. Seaweed has been studied as an ingredient to include in processed meat products as a way of reducing the amount of salt normally added to such products while adding additional benefits to the consumer such as increased plant protein intake [87]. As a naturally salty plant, seaweed can be used as a salt substitute while also providing the consumer with important minerals like potassium and magnesium [88]. Seaweeds are deemed “healthier” as a salt alternative when they have low Na:K ratios, as is seen in *Laminaria digitata* and *Palmaria palmata* species (which have ratios of 1.03 and 0.84 Na:K, respectively) [89]. These ratios vary from species to species and therefore seaweed as a general product cannot be assumed to be a safe salt replacement, where in some instances the addition of a 5 g portion of seaweed can provide more salt and sodium than an equivalent portion of bacon [16].

As a low-sodium salt ingredient, red seaweed, which is higher in potassium and magnesium than regular salt, was approved by a panel of sensory testers and therefore the authors propose the use of this seaweed as a safe alternative to salt [90]. In a study determining the risk associated with seaweed consumption and adverse cardiovascular events, a reduced risk of mortality in those people who consumed seaweed was found [91]. However, the authors declared that seaweed intake may be a reflection of a healthier person and that other factors may be contributing to this decreased risk [91]. Possible mechanisms behind the studied reduction of risk in cardiovascular diseases and seaweed consumption have been shown in animal studies through the lowering of blood pressure [75], and a serum lipid controlling effect [92]. The presence of salt in seaweed can also have an impact on its processing strategies (Section 3.3.3).

## 3. The Processes

### 3.1. Post-Harvest Technologies for End-Product Optimization

Due to their rich nutritional profile, as discussed in the previous section, and high moisture content due to their natural habitat, seaweeds are a suitable substrate for microbial growth. It is therefore very important for seaweed aquaculture farmers to find ways to minimize the risk of pathogenic microorganisms by adopting postharvest techniques that allow for the preservation of the seaweed biomass while also preserving important nutrients and increasing the shelf life of the seaweed. The most common postharvest treatments for seaweed include cleaning, soaking, blanching, and drying/dehydration [93].

Cleaning or washing of the seaweed is normally carried out directly after harvesting to remove any contaminants such as sand or crustaceans. Washing and soaking as a pre-treatment step were found to lower the potentially damaging toxin arsenic [94] while soaking (1–24 h in deionized water) and boiling were two methods used to successfully lower iodine levels in *A. esculenta, P. palmata*, and *U. intestinalis* [95].

Blanching is a common measure employed in food processing to deactivate enzymes associated with product degradation [42] and decrease subsequent drying times [96]. Cleaning, soaking, and blanching can be carried out with minimal onsite equipment and at very little cost. Simple materials are required for the process; clean, food-grade containers (preferably stainless steel) that have a connection to a potable water source. Blanching can also be carried out using a water-bath [97]. Blanching can be employed to reduce the presence of potential toxins like microbes [98], heavy metal content [99], and high-levels of iodine [97].

Drying is usually the next step in seaweed postharvest methods, and the final stage before the biomass is directed for further processing depending on its end use. Where climate allows, solar drying is commonly used and is a cheap and efficient way to dry large amounts of seaweed in a short space of time [100]. Conventional hot air drying is used by commercial seaweed farmers to dry large quantities of seaweed, however, this is associated with high-energy costs. There are multiple alternative drying technologies such as microwave drying, dehumidified air-assisted drying, vacuum drying, freeze-drying, and ultrasound-assisted fluidized bed drying [93]. Table 4 highlights the impact of different drying methods and temperatures on seaweed quality and energy efficiency. Different drying conditions like temperature, time, and method can have a significant impact on the end-product applications. Seaweed producers need to be aware of the impact of the particular drying method they use when selecting equipment, as it may negatively impact certain aspects of their end-product quality.

Novel technologies are emerging for maximizing the efficiency of the postharvest stage, such as microwave and ultrasound-assisted blanching and drying technologies such as freeze-drying [93]. Novel blanching methods were more effective at retaining more volatile compounds, when compared to conventional blanching [37]. When examining the impact of drying method on the volatile profile of *A. esculenta*, freeze-drying retained more volatile compounds (quantity) while oven-drying retained a more diverse profile of volatile compounds (quality) [37]. The direction of the end product will influence the seaweed manufacturers’ choices in processing methods, however, there is no doubt that these postharvest strategies are a crucial stage in the seaweed food production industry. Traditionally speaking, seaweed food products are made using minimal processing. With an increase in both production and demand for seaweed worldwide, processing methods are being adapted for seaweed novel food product development.

### 3.2. Milling for Seaweed Powder

In traditional flour milling, the chosen method of milling has impacts on flour quality, dough rheological properties, and end-product bread characteristics [111]. The priorities of the company producing the bread or dough product will determine the method of milling employed, as highlighted by Capelli et al. 2020 [111]. In this review, the advantages and disadvantages of these two popular milling methods are examined. Stone milling is easy to use and produces a flour with “higher concentrations of macroelements, microelements and polyphenols”, whereas roller milling is more efficient and flexible with a lower heat production and produces a dough with improved rheological properties [111]. In this way, the end-product purpose should be considered when selecting the milling technique to use in seaweed processing.

Milling seaweeds can be a challenging process, particularly brown species which have a higher fibre content and can therefore be tougher to grind. The type of mill used is an important factor in determining seaweed powder quality and consistency. Ball milling, hammer milling and pulverizing techniques are all suitable ways of grinding dried seaweed into a powder form. One such process is described by a patent for improved seaweed milling whereby a two-step process using a hammer mill for a short period, followed by a separation step (to remove particles of a certain size) is employed to develop a fine seaweed powder of the desired particle size, as determined by an air classifier [112]. In this design, the polyphenol content and antioxidant activity were found to be higher when compared with other milling methods, such as hammer milling [112].

Particle size is an important feature to consider when choosing a milling method to produce flour. Many dough properties are affected by flour particle size; such as pasting properties, hardness, and springiness [113]. Protein content, amino acid profile, and other nutritional components can become disrupted through milling and therefore the various grades of powders produced from the process may contain different levels of these molecules [9]. Therefore, seaweed producers need to assess the individual nutritional profile of each powder grade produced.

### 3.3. Green and Novel Technologies for Seaweed Processing

Different food processing technologies are employed based on different end-product applications. For instance, size reduction processes such as milling, slicing, pulping, or homogenization, may be used to change their organoleptic characteristics and allow for further processing [114]. Physical separation is sometimes necessary to produce juices and oils and involves processes like centrifugation, filtration, or membrane separation, among other methods [115]. In seaweed processing, this can occur in the format of sieve separation into fractions of different particle sizes for dried seaweed or through other mechanical methods like extrusion or hydrodynamic cavitation in the case of the separation of liquid and solid fractions of seaweed biomass. Such processes are regarded as energy-efficient processing methods due to shorter processing times.

#### 3.3.1. Extrusion

Extrusion is a process used to obtain a range of food products and occurs when food ingredients are forced through conditions of mixing, heating, and shear, through a die that forms and/or puff dries the ingredients [116]. During the process, microorganisms and enzymes are inactivated while the nutritional quality of the material is retained [117]. It is a continuous process where the material is cooked under pressure, moisture, and elevated temperature to create textured foods such as snacks and breakfast cereals. Seaweed is a suitable substrate for extrusion. Given the rich nutritional profile of seaweed, it is an ingredient much desired by the food industry for inclusion to fortify or enrich products such as puffed snacks or chips. The bioactive profile of extruded food products is affected by extrusion process variables such as shear, temperature, resonance time, and water content [118]. *Sargassum tenerrimum* was extruded for nutritional and quality analysis with results showing valuable protein and lipid profiles contained in the extruded seaweed product [119]. Physical properties such as expansion ratio and textural properties such as porosity and hardness were influenced by seaweed inclusion rate [119]. An increase in *S. tenerrimum* content and feed moisture level in the fortified product caused a decrease in crispness and an increase in hardness, which may be due to the high fibre content of seaweeds [119]. The unique nutritional profile of different seaweed species may need to be considered by food processors using seaweed as an ingredient in enhanced products as a result. Optimal extrusion measures for alginate extraction from brown seaweed using a twin screw extruder were evaluated by Singh et al. (2018). Fixed optimized parameters of an algae to solution ratio of 3.11:1, feed rate of 2.95 rpm, and a pH of 10.3 produced the most desired results, an RTD (residence time distribution) of 6.80 ± 0.089 min, yield (ratio of extracted alginate weight to initial weight of dry seaweed) of 34.96 ± 0.09%, and alginate molecular weight 211.93 ± 8.74 kDa) [120]. Based on these results, the authors conclude that a twin screw extrusion process is a very promising method of alginate extraction from brown seaweed [120].

Reactive extrusion involves the feeding of reactants through the main hopper and into the twin-screw extruder where heating occurs to start the chemical reaction [121]. This process can be used to efficiently extract alginate from seaweed [122]. Hydrocolloid extraction from seaweeds is a large industry. The alginate market was valued at $728.4 million in 2020, while carrageenan was valued at $741.9 million in 2019, with a CAGR (compound annual growth rate) of 5% from 2021 to 2028 and 5.9% from 2020 to 2027, respectively [123,124]. Current industry extraction methods require a lot of energy and chemicals [61]. Nonchemical extraction methods for the purpose of hydrocolloid extraction include microwave-assisted extraction, ultrasound-assisted extraction, enzyme-assisted extraction, supercritical fluid extraction, pressurized solvent extraction, and reactive extrusion, (Khalil et al., 2018) [61]. These promising technologies offer new methods of extracting seaweed functional hydrocolloids in a sustainable, more efficient, and environmentally friendly way. However, there are some drawbacks to these technologies such as the high cost of equipment (microwave-assisted extraction and supercritical fluid extraction) or slow extraction process (such as enzyme-assisted extraction) [61].

#### 3.3.2. High Pressure Processing

High pressure processing (HPP) involves exposing foods to high pressures (300–700 MPa) for a short period of time (seconds to several minutes) [125]. The process inactivates microorganisms and enzymes to extend the product shelf-life without the need to apply heat [126]. While not considered a “novel technology”, the application of technologies like HPP to a new biomass like seaweed is evolving and novel products are being generated as a result. The benefits of utilizing these technologies for seaweed are many, including increasing shelf life, improving manufacturing efficiency, and rapid new product development, which allows seaweed aquaculture farmers to upscale quickly and expands the potential markets they can enter with their products. Olmo et al. (2020) evaluated five species of seaweed for preservation by HPP [127]. HPP-treated samples (600 MPa for 5 min) had a shelf life greater than 180 days for all five seaweed species tested, compared with a range of just 3–60 days among non-HPP treated controls (different values for different species) [127]. They concluded that the total phenolic content (TPC) was not affected by HPP, while seaweed species and days kept in storage showed significant influencing factors on TPC [127].

#### 3.3.3. Saltwater Based Fractionation of Seaweed for Biorefinery

As well as implications for certain consumer groups (such as those predisposed to cardiovascular issues), the presence of salt in seaweed can have an impact on processing, creating a need for potentially energy-intensive stages such as washing and drying [128]. Salt can cause issues relating to the effectiveness of downstream processing and engineering infrastructure needed [128]. For biological processing, such as fermentation, salt must be removed in order to support the microbial populations used in this system [129]. Jones, et al. (2020) propose a biorefinery approach to solving the salt problem of macroalgal biomass processing using a salt-based biochemical conversion route and revalorizing the aqueous phase for fermentation by a salt tolerant yeast species [128]. This approach could be implemented for large-scale conversion of seaweed into biofuel or fertilizer products.

#### 3.3.4. Seaweed Fermentation

Humans have a long history with fermentation, first born out of a necessity to preserve food in periods of low harvest/foraging yields. The most popular fermented products such as wine, beer, and dairy were first developed as far back as 13,000 BC [130]. Since then, we have continued fermenting a wide range of nutritional sources while the methods have remained fairly stable as technology has developed. Many countries and cultures have a strong connection to traditional fermented goods from their geographical region [131]. Fermentation of seaweed is an underused biotechnology tool that can be used to extract valuable bioactive molecules and develop novel products [132]. Fermentation is a well-established method of food preservation and processing that has been employed for centuries by humans. Seaweed fermentation can be used as a technology to develop pharmaceuticals and health supplements, as well as medical products [133,134]. Fermentation is an energy-efficient process and therefore more environmentally sustainable than other processing methods such as chemical extraction, which can produce a lot of waste [132]. Under the process of fermentation, many biochemical interactions are occurring, finally resulting in the production of secondary metabolites [135]. In seaweed fermentation, hydrolysis is employed to first break down the main structural polysaccharides so that glycolysis can then be used to convert these sugars to pyruvate [132]. This pyruvate can then be fermented through alcoholic or lactic acid fermentation to produce ethanol and CO_2_ [136]. Further extraction or purification is then employed to access polyphenols, bioactive amino acids such as threonine and leucine or flavour-enhancing amino acids like glutamic acid [136].

## 4. The Products

While there are many food applications for seaweed that have been around for thousands of years, tailoring them to the contemporary consumer requires a rethinking of the use of seaweed in our current food systems. New technologies, as discussed previously, allow for the new product development of very traditional recipes like seaweed bread. The nutritional and functional properties of seaweed allow it to be examined for a range of food products for human consumption. Furthermore, foods enriched with seaweeds have added health benefits for the consumer, making them value-added products, and a very interesting area to study in new food product development. The unique properties of seaweeds can open up many possibilities for food product development with this focus in mind. For instance, seaweed can be used as a salt replacement in meat products. New product development is a key part of the business growth model of a company. Without it, the company is stagnant. As the market evolves, so too must the products and innovations to keep up with consumer demand. Table 5 highlights the impact of adding seaweed to various food products such as noodles, bread, jelly, and meat products.

### 4.1. Seaweed Powder for Dough Products

Flour is a staple in many human diets worldwide, although its application and the source of raw materials (i.e., grain) vary greatly. Flour is produced by grinding a carbohydrate source (such as grains, roots, beans, nuts, or seeds) to make a fine powder. It may also be fractionated to obtain optimum particle size for incorporation into the dough.

Some nations already have a long-standing tradition of incorporating seaweed in breads. In Germany and Austria, a bread called algenbrot is made with a blend of cereals and up to 3% seaweed, while in Brittany, dulse and kombu variations are used to make *Bara mor* (“bread of the sea”) [147]. Bread consumption is widespread throughout all nations in some form or another, with over nine billion kg of bread produced in the world annually [148]. The inclusion of other ingredients into the dough is becoming more popular, as the demand for healthier bread increases. The addition of seaweed into bakery products such as bread dough not only improves the nutritional quality of the food product, it can also enhance the techno-functional properties of the bread. Mamat et al. (2013) reported that the addition of powder from the red seaweed species *Kappaphycus alvarezii* (at a rate of 2–8%) increased water absorption of the dough, decreased stickiness and showed improved firmness [64]. Such findings have implications for food producers as they are urged to look for more natural, sustainable methods of preserving and improving dough quality. Rico et al. (2018) developed novel flour ingredients to create value-added bread products using seven seaweed species for comparison with carob (a fruit used to produce locust bean gum for the food industry) byproducts for suitability of incorporation to dough [149]. They found that, while seaweed is a rich source of phenolic acids displaying antioxidant functionality, carob flour had a higher antioxidant activity than seaweed flours “probably associated with non-extractable phenolic compounds linked with fibre” of the seaweed [149]. For this reason, extraction processes as mentioned in Section 3 will be critical in ensuring seaweed is able to meet the requirements for incorporation into products like functional doughs.

Amoriello et al. (2021), found that the inclusion of a number of different algae sources had positive impacts on bread characteristics, however, they were dependent on the type of species and inclusion rate [150]. Seaweed inclusion had a darkening effect with a shift towards the green and blue spectrum [150]. Crumb porosity was improved by the addition of kombu (*Laminariales* spp.) and sea lettuce (*Ulva* spp.), as well as an improvement in the total soluble phenolic compounds, pigments, and antioxidant activity profile of the bread [150]. Inclusion of *A. nodosum* and *C. crispus* had a positive impact on the mineral profile and dietary fibre content of the end product when added to whole-wheat bread [73]. However, inclusion rates of 4% and 2% for the two species, respectively, were optimum due to adverse sensory impacts above these thresholds [73]. Other wheat-based products such as pasta showed an improved amino acid and fatty acid profile of the finished pasta product when *Undaria pinnatifida* was added to the recipe and was successful in passing the sensory screening when included at a rate of 10% [151].

### 4.2. Seaweed Snacks: Convenience Is Key to the Modern Consumer

Nutritionally, the addition of seaweed to extruded products (often thought of as unhealthy “snack” products) can enhance the overall profile of the product and promote the consumption of seaweed in everyday life [119]. The natural saltiness flavour profile of seaweed makes it a prime candidate as a salt alternative, with added benefits of the nutritional factors outlined in Section 2. When incorporated into a spice mix at a rate of up to 20%, the product had a high consumer acceptability, with an enhanced protein, fibre, and mineral content [146]. This saltiness profile makes it a suitable ingredient to incorporate into fermented products. Successful fermentation of kelp with lactic-acid bacteria was carried out and improved the phenolic content and antioxidant activity of the sauerkraut-like end products [152].

An opportunity to develop sweet or sweet–savory snacks in the Japanese market was identified by Kumar, 2021 #236, in an assessment on market demand and potential areas for exploitation by the food industry. This study also identifies creative ways to develop new products such as increasing the plant protein content of existing products to deliver a higher-protein product to the consumer within an established market space [153]. In this way, popular texture markets (crispy, dry snacks) are targeted, though with an innovative component (“environmentally sustainable ingredients”, “plant protein”, “natural”) [153]. Descriptive analysis (texture and flavour profiles) is key to gathering information on sensory preferences and help guide innovators towards the best markets for new product development.

Extrusion, as discussed in Section 3.3.1, is a method by which ingredients can be processed on a large-scale to make convenient snack foods. Extruded foods can “provide nutritious products and combine quality ingredients and nutrients” to make food products that are made up of exact proportions of each ingredient [154]. When included as an ingredient in a rye-based extruded snack, seaweed (*Fucus vesiculosus*) was found to have positive nutritional benefits to the food through the addition of natural antioxidants and improving the oxidative stability of the snack product [155]. The extrusion process did not impact the antioxidant activity of the seaweed within the snack end product [155]. The inclusion of seaweed in extruded snack products is a welcome innovation to fortify the product and increase its nutritional quality, thus improving the overall quality of the snack food [119]. However, inclusion rates of seaweed need to be carefully selected by manufacturers as increased rates (over 7.5%) are associated with a lowered acceptability score through sensory testing [119].

### 4.3. Seaweed as a Supplementation in Dairy Products

Seaweed extracts from *A. nodosum* and *F. vesiculosus* at low inclusion rates (<1%) were found to be stable in milk and improved the antioxidant activity (DPPH and FICA) of the product in varying degrees [145]. Historically, seaweed such as *Chondus cripus*, or carrageen moss, has been consumed by boiling in milk to create a jelly-like dessert [147]. dulse, duileasc (Gaelic), or *P. palmata*, was commonly consumed in Northern European countries alongside dishes of “dried fish and butter or with milk and bread” [147]. In Northern Ireland, dulse champ is a dish of mashed potatoes with milk, butter, and cooked dulse as seasoning [147]. As a thickening agent seaweed has been used in sauces and soups, with its inclusion in Indonesian cooking to enhance the thickening properties of coconut milk [156]. Today, carrageenan is a widespread ingredient in the food industry to make dairy gels [157] and to stabilize products such as ice cream [158]. As a plant source, carrageenan is a valuable alternative to gelatin for the vegan market. O’Sullivan et al. (2016) found that seaweed added to yoghurt had no impact on pH, microbiology, and whey separation of the yoghurt product and that some treatments experienced lower levels of lipid oxidation [159]. Seaweed can be used as an ingredient to create novel fermented milks, with results showing probiotic bacteria growth stimulation in milks with added seaweed (species dependent) [160]. Seaweed can be added to cheese as a means of enhancing end product nutritional parameters. In a study where 10 g of dried seaweed was added for every 1 kg of curd, there was a significantly higher rate of antioxidant activity (correlated with the phenolic profile of the seaweed) when compared with the control cheese [161]. Dry matter and pH value were affected by the inclusion of seaweed, however, microbiota was not significantly affected [161].

### 4.4. Seaweed for Meat and Fishery Products

Apart from textural, stabilizing, and gelling functional properties, as mentioned in previous sections, seaweed can offer other functions to food such as prepared seafood products. When included in fish surimi (an East Asian fish paste product), the edible green seaweed *Ulva intestinalis* has a positive impact by having a lower TBARS (thiobarbituric acid reactive substances, a measure of lipid peroxidation) value when compared with the control, showing a potential use in product quality and shelf-life extension [162].

Seaweed extracts were included in minced tilapia preservation as a natural alternative to synthetic preservatives and the resulting product successfully met regulatory quality standards including microbiological limits [163]. In canned salmon, seaweed was incorporated as a natural means of preserving the product [164]. Due to the natural antioxidants within the seaweed extracts, the addition of seaweed had some positive improvements to the salmon product such as a decrease in secondary peroxidation when compared to the control sample, a higher PUFA content, and lower oxidized odours with no impact on sensory quality parameters [164].

The demand for healthier meat products is increasing due to consumer demand for higher quality meat parameters, one of which is lower salt content. Seaweed can enhance the nutritional profile of meat by increasing the omega-3 content, decreasing the omega 6:omega 3 ratio and decreasing the TI (thrombogenic index), a measure used to determine the impact of food on heart health [165]. Meat products made with seaweed had improved sodium profiles as well as a better mineral profile (specifically for K, Ca, Mg, and Mn) and amino acid profile (serine, glycine, alanine, valine, tyrosine, phenylalanine, and arginine), depending on the species tested (wakame and sea spaghetti saw no significant impact on the amino acid profile of the meat product) [165]. Pork liver pâté was enhanced by the addition of seaweed, with an increase in protein content of 2–3%, and a similar degree of protection against oxidation when compared to a synthetic antioxidant compound (BHT or *Butylated hydroxytoluene*) [166]. Such results demonstrate the unique ability of seaweed to not only enhance the nutritional profile of food and they offer a natural preservation benefit through antioxidant activity. Seaweed can be used as an ingredient in other meat products such as patties and frankfurters as a means of reducing sodium and saturated fatty acid content of the products, while simultaneously adding minerals, omega-3 fatty acids, and polyphenols, therefore offering the consumer a healthier option than the original product [167].

### 4.5. Seaweed for Gluten-Free Products

Due to their hydrocolloid content and known functional properties, as discussed in Section 2.3, seaweed can be used as a textural aid ingredient in gluten-free products. When added to a mixture of gluten-free pasta dough and compared with a gluten-free pasta control, the seaweed-containing pasta presented similar textural properties to the control, though with significantly higher fibre and mineral content [143]. The addition of *K. alvarezii* to a gluten-free pasta mixture was found to improve the overall quality of the pasta through an increased viscous elasticity, higher calcium and dietary fibre content, improved cooking properties, and greater consumer preference for gluten-free pasta when compared with a gluten-free control pasta [168].

Vestå, 2022 #585 reported that the inclusion of the seaweed hydrocolloid alginate in a gluten-free bread mixture was successful in terms of textural quality, however, consumer acceptability of the product was lower than control samples.

### 4.6. Seaweed Gastronomy

Japan is probably the most well-known seaweed-consuming nation in the world. However, many other countries and cultures have long-standing traditions with seaweed consumption. Island nations such as Ireland and Iceland have seaweed recipes that have been handed down through generations. Such records may see a revival in the coming years as Western countries seek a renewal of seaweed in their diet. Modern takes on traditional ingredients is a popular trend in the culinary industry, as was seen in a collaboration between Michelin-starred chef Koji Shimomura and scientist Hiroya Kawasaki who designed a new seaweed dish based on nutritional value and taste [169]. Rioux (2017) reviewed seaweeds as an ingredient to be utilized by the food industry as a traditional ingredient for a “new gastronomic sensation”, recognizing its potential to deliver valuable bioactive nutrients to the consumer, as well as using seaweed as “a vector of flavour and texture” [170]. Perhaps overlooked is the fact that seaweed is already a very common ingredient in our foods and many of us unknowingly consume it as an ingredient on a daily basis as a textural agent. Considering the large market size of these hydrocolloids, as mentioned in Section 2.3, it is clear there is a lack of research in the area of the physicochemical properties of seaweed and its impact on gastronomy, and the potential applications in novel food innovation, as highlighted by Mourisen (2012) [171]. Here, the scope of seaweed and its use in gastronomy is reviewed, as well as the “unexplored” areas of potential research, such as an assessment of the biophysical properties of omega-3 fatty acids found in seaweed or the definition of the physics of the taste sensory system stimulated by glutamate present in seaweed [171]. As discussed, seaweeds are exploited for a number of physicochemical properties to enhance a wide range of food and other ingredients. Molecular gastronomy is a branch of the food industry which focuses on applying scientific principles to produce innovative products or dishes [172]. Examples of seaweed in molecular gastronomy include the use of alginate to produce spherical features with thin membranes enclosing a liquid or agar to make “spaghetti” or “caviar” type gels [173].

### 4.7. Nonfood Seaweed Products

#### 4.7.1. Smart Packaging—Technologies to Produce Sustainable, Green Packaging Solutions

As the consumer becomes more environmentally conscious, pressure is put on supermarket retailers to make more sustainable choices when sourcing suppliers to stock in their shops. This not only relates to the sustainability of the food products themselves but perhaps as importantly, to the packaging they come in as well. Proper packaging is a crucial part of the food production system as it allows for extended shelf life and a convenience of handling for both the consumer and supply chain logistics personnel. However, an unfortunate byproduct of this system is the build-up of enormous amounts of packaging waste, most of which is plastic-based. To combat this, recycling is encouraged, however, ultimately, biodegradable or biobased alternatives need to be developed to replace plastic waste. The difference between biodegradable and biobased plastics is the source from which they are manufactured [174]. Carina et al. (2021) described biodegradable polymers as being made from “natural and fossil resources” while biobased plastics are made up of 100% extracts “from plants, marine organisms or produced by microorganisms through fermentation” [174]. Smart—or active—packaging is defined by *Carina* et al. (2021) as “a system in which product, packaging and environment interact in a positive way to extend the shelf life, improve the safety as well as the sensory properties and maintain the quality of the product” [174].

There are a number of different types of active components in packaging such as oxygen scavengers, carbon dioxide emitters, antimicrobial agents, moisture control, odour absorbance and ethylene scavengers [174]. The use of such chemicals in the packaging in direct contact with food for human consumption could be a cause for concern due to the potential toxicity of these agents or the nanocomposites [175,176,177,178,179,180]. As a result, there is interest in developing natural compounds for such a purpose that would be safe to consume by humans.

Polysaccharides extracted from seaweeds are being proposed as valuable sources of such novel packaging [174]. Seaweed polysaccharides have a range of well-studied functional properties such as antioxidant activity, antimicrobial activity, photoprotective activity [181], and thickening, gelling and stabilizing properties [6] that increase its potential as a novel, sustainable packaging source. Furthermore, its antimicrobial properties, such as those in *H. elongate* have been shown to inhibit the growth of gram-positive and gram-negative bacteria, giving it added value for use as a protective packaging material and thus extending shelf life [174]. Doh et al. (2020) developed seaweed nanocomposite films from two brown seaweed species, *Lamianria japonica* and *Sargassum natans* [182]. When reinforced with CNCs (cellulose nanocrystals) extracted from the seaweeds, the tensile strength of the films, as well as the water, oxygen and light barrier properties were significantly improved when compared with the films without CNCs [182].

#### 4.7.2. Seaweed in Cosmetics

Mycosporine-like amino acids are a group of secondary metabolites found in seaweeds, fungi, and microorganisms like cyanobacteria and microalgae [183]. These antioxidant, UV-absorbing molecules are produced in seaweeds when exposed to high levels of UV-A and UV-B [184]. Lawrence et al. (2018) evaluated that palythine, an MAA extracted from seaweed, was protective against UV while also offering antioxidant benefits, even when added after sun exposure [185]. As a natural active ingredient, seaweed is a valuable ingredient for skincare products such as face masks, where the seaweed also offers the technical ability to form a “thermos-reversible gel that can stabilize the mask preparation” [186]. Sulfated polysaccharides extracted from *Ecklonia maxima* were found to reduce the impact of skin damage, as well as inhibit wrinkle-related enzymes and improve collagen synthesis (when tested on UVB-irradiated human dermal fibroblasts) [187]. The researchers determined that the polysaccharides tested have a strong potential to be used in antimelanogenesis and photoprotective activities, and should be targeted as an ideal ingredient for the cosmetic industry [187].

## 5. Conclusions

### 5.1. Economic Importance of Seaweed

As mentioned earlier, seaweed is unique in its farming system in that minimal inputs are required, unlike other forms of agriculture where fertilisers, feed, and pesticides are employed to ensure adequate yield. This feature of seaweed aquaculture allows poorer communities to adopt the enterprise more readily than other forms of farming [188]. In some countries, like Tanzania, seaweed aquaculture has replaced other industries like fishing and subsistence farming with an observed improvement in the economic status of the people who changed industries [189]. One of the many positive findings from this study was increased retention of young people in the villages, who chose to stay to work on seaweed farms instead of moving to towns [189]. In Indonesia, seaweed farming is attractive to people as a business for the reasons already mentioned and has the ability to provide families with a regular income, and it can even bring rural households above the national poverty line [190]. Such examples are observed in rural coastal communities worldwide and provide positive support for seaweed aquaculture. These farms are usually selling the raw material seaweed, and, as the seaweed is further processed, the value increases [188]. There are further opportunities for these farms to include some processing so that they can increase their business reach and develop longer value chains themselves. The Philippines have the most complete value chain, as reviewed by the FAO, in terms of carrageenan production while India has the shortest value chain (meaning most farmers sell directly to the processors) [188].

### 5.2. Strengths and Opportunities

Seaweed is a promising source of nutrients for humans. Blue biomass can offer consumers a way of increasing their plant protein consumption and expanding their mineral intake, particularly iodine. For the food industry, seaweed is a valuable source of functional biomolecules like alginate, agar, and carrageenan. For the nutraceutical industry, bioactives like polyphenols and carotenoids can be extracted and produced in concentrated form to deliver health benefits to the consumer. Seaweed has been a source of nutrition for humans throughout our history. How our relationship develops in the coming generations will depend on market demand and research into the wide range of species of seaweed, their specific optimal environmental conditions, processing methods, and extraction techniques. The range of products seaweed can be incorporated into is vast and there are multiple lines suitable for targeting immediately, such as the healthy meat sector. Seaweed gastronomy is an innovative avenue for seaweed researchers, producers, and culinary partners to collaborate and explore the potential of seaweed in food. Environmentally, seaweed aquaculture is an attractive model with no need for land use, fertilizers, or fresh water.

### 5.3. Challenges and Aspirations

As with any food system, scaling up for large commercial production comes with challenges. Today, there is a focus on developing sustainable systems that not only cause minimal damage to the environment and actually help solve an environmental problem. Seaweed is one such sustainable system being proposed, however, processing choices will be key to ensuring that the environmental impact is minimized [191].

Increased information campaigns surrounding the benefits of consuming seaweed, as well as the positive impacts on the environment, will help overcome the challenges associated with neophobia. Defining the environmental impacts of large-scale seaweed aquaculture needs to be carried out in order to provide reliable data on the sustainability of the system based on real-life metrics and not models based on small-scale production. Determining end product quality will be a challenge for seaweed producers due to dynamic variables in its natural environment. It is difficult to assess seaweed in general terms as the variation in nutritional profiles varies so greatly across species and even seasonally within species, as is evident from this literature review. There are some hurdles regarding seaweed processing such as the financial cost of expensive drying systems and its impact on energy consumption and the environment. More efficient processing technologies such as extrusion can aid in making seaweed processing a greener food production system.

The European Commission recently released a report entitled “Blue Bioeconomy—towards a strong and sustainable EU algae sector” outlining the opportunities and challenges faced by the seaweed aquaculture industry [192]. As highlighted in the document, “algae-based food is not traditionally consumed in the EU” and they acknowledge the need to create market awareness of the benefits of consuming seaweed both for the nutritional and environmental benefits [192]. According to this document, there are 1550 different seaweed species growing in EU waters, however, only 23 have been included in the food or food supplements of the EU Novel Food Catalogue [193]. Some of the most popular species farmed and sold on the European market are *Saccharina latissima*, *Alaria esculenta*, and *Ulva* sp. [192].

Observing the global demand for more clean and green sources of food with beneficial qualities for the consumer demonstrates the need for seaweed aquaculture to overcome these industry challenges and become a valuable player in the search for new and improved food systems.

## Figures and Tables

**Figure 1 biomolecules-13-00386-f001:**
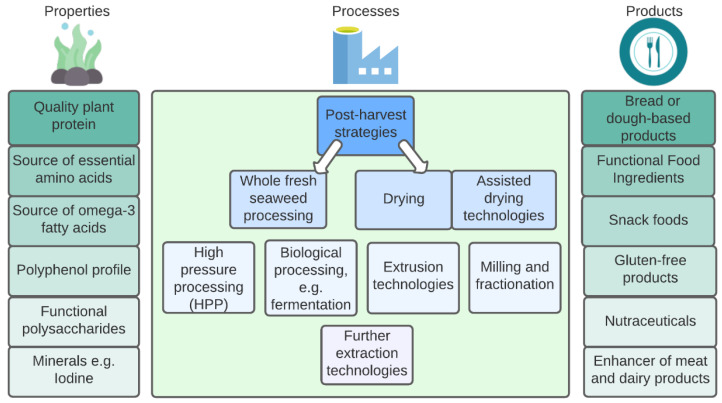
Scheme of seaweed properties, seaweed aquaculture processing and resulting products.

**Table 1 biomolecules-13-00386-t001:** Protein content of some red, green, and brown seaweeds on the Swedish west coast, adapted with permission from [8]. 2020, Olsson et al.

Seaweed Phylum	Species	Protein Content (g kg^−1^ dw)
*Chlorophyta*	*Cladophora rupestris*	184
	*Ulva intestinalis*	90
	*Ulva lactuca*	93
*Rhodophyta* (Red)	*Ahnfeltia plicata*	201
	*Chondrus crispus*	103
	*Delesseria sanguinea*	183
*Phaeophyceae* (Brown)	*Ascophyllum nodosum*	59
	*Fucus serratus*	71
	*Laminaria digitata*	66

**Table 2 biomolecules-13-00386-t002:** The top five amino acids (mg g^−1^) found in each of the three seaweed species studied [9].

Species	Glutamic Acid	Aspartic Acid	Glycine	Alanine	Leucine
*Alaria esculenta*	55.34	40.48	36.22	35.64	24.69
*Laminaria digitata*	23.78	15.42	14.54	13.42	12.78
*Saccharina latissima*	18.40	19.41	10.59	9.98	12.55

**Table 3 biomolecules-13-00386-t003:** Sources of hydrocolloids and the products derived from them. Adapted and sourced from: www.cybercolloids.net (accessed on: 5 November 2021) [63].

Raw Material	Key Products
Tree exudates	Gum Arabic, Tragacanth, Karaya
Seed flours	Guar Gum, Locust Bean Gum, Tara, Cassia Tora
Plant fragments	Pectin, Cellulose
Fermentation biomass	Xanthan, Curdlan, Gellan
Seaweed extracts	Carrageenan, Agar, Alginate
Animal origin	Gelatin, Chitosan, Isinglass

**Table 4 biomolecules-13-00386-t004:** An overview of various drying technologies used in seaweed processing, adapted with permission from [93]. 2021 Zhu et al.

Drying Method	Conditions	Seaweed Species	Final Moisture Content (MC)/Ratio (MR)	Results	Drying Kinetics Model	Reference
Dehumidified air assisted tray drying	T: 40 to 70 °C; V: 5 and 7 m/s; DT: 100 to 3000 min	*Eucheuma cottonii*	MC: 15% (w.b.)	Higher air temperature and air velocity resulted in faster water removal. Moreover, temperatures below 70 °C resulted in a reasonable seaweed quality	Page model	[101]
Solar drying and shade drying	Solar DT: 5 days; Shade DT: 8 days;	N/A	MC Solar: 24–61% (d.b.); MC shade: 40–48% (d.b.)	Samples dried unevenly. Henderson and Pabis model was adopted	Henderson and Pabis model	[100]
Hot air drying	T: 35 to 75 °C; RH: 30%; V: 2 m/s; DT: 120 to 240 min	*Ascophylum nodosum*, *Undaria pinnatifida*	MR: 0.03	Temperature affected drying time and color significantly. Conventional air drying can be considered adequate for *A. nodosum*, but not for *U. pinnatifida.*	Page model	[102]
Osmotic dehydration assisted hot air drying	T: 30 °C; RH: 14%; DT: 2 h	*Porphyra columbina*	MC: 7.9% (d.b.)	Osmotic dehydration, as a pretreatment for air-dried seaweeds, did not seem to improve the final product quality	Page model	[103]
Vacuum drying	T: 40–80 °C; P: 15 kPa; DT: 180 to 800 min	*Pyropia orbicularis*	MR < 0.1	Vacuum drying at 70 °C had the highest total phenolic, carotenoid and phycoerythrin and phycocyanin content, lightness as well as antioxidant capacity.	Weibull model	[104]
Sauna treatment assisted solar drying	T: 35–40 °C; RH: 32–80%; DT: 2 days	*Kappaphyccus alvarezii*	MC: 35% (d.b.)	Sauna treated seaweed reduced the drying time by 57.9%	Page model	[105]
Spray drying	T: 140–180 °C; FFR: 3–5 rpm;	*Sargassum muticum*	MC: 1.83–3.83% (d.b.)	Good-quality, stable seaweed powder with acceptable properties was spray dried at 140 °C and 3 rpm, with 4% of maltodextrin.	N/A	[106]
Freeze drying	T: −86 °C; DT: 48 h	*Kappaphycus alvarezii*	MC: 11% (d.b.)	Freeze drying did not show any benefit to retaining any seaweed chemical compositions	N/A	[107]
Ultrasound assisted fluidized bed drying	US: Fre: 26 kHz; P: 170 W; V: 6.7 m/s; DT: 110 min USP: Fre: 20 kHz; P: 500 W; DT: 80 min	*Ascophylum nodosum*	MC: 10% (d.b.)	Airborne ultrasound dried recovered the best total phenolic content as well as colour, however, no benefit in reducing drying time. Ultrasound pretreatment had the lowest drying energy consumption.	Page model	[108]
Fluidized bed drying	T: 40–60 °C; V: 0.5–1 m/s	*Echium amoenum*	N/A	The optimal drying conditions were air velocity of 0.86 m/s at 60 °C in terms of highest bioactive compound content, and minimum drying time.	N/A	[109]
Spray drying	Pretreated with USP T: inlet 175 °C/outlet 80 °C	*Gracilaria secundata* combined with amaranth protein	N/A	Spray drying can be used as an alternative to freeze-drying when producing conjugates with observed improvement in water holding capacity.	N/A	[110]

Note: T: drying temperature; V: air velocity; DT: drying time; FFR: feed flow rate; *U.*: *Ulva rigida*; P: power; Fre: frequency; US: airborne ultrasound assisted fluidized bed drying; USP: ultrasound pretreatment; MC: moisture content; MR: moisture ratio; N/A: not applicable.

**Table 5 biomolecules-13-00386-t005:** Impact of seaweeds addition on food properties, adapted with permission from [93]. 2021 Zhu et al.

Food Product	Seaweed Species	Seaweed Processing	Impact	**Reference**
Jelly	*Gracilaria verricosa*, *Ulva lactuca* & *Sargussum wightti*	Washing Extraction from fresh biomass Seaweed extracts employed: 0.1, 0.5, 1.6 mg, 200 mL^−1^	Seaweed colours retained more than thirty days at room temperature and 30% loss similar to artificial colours. Seaweed colours contributed higher nutrition content in final product.	[137]
Pork	*Laminaria digitata &* *Fucus vesiculosus*	Pre-treatment not applicable Seaweed extracts employed: 100 mg g^−1^ pork	Laminarin had no antioxidant activity but fucoidan reduced lipid oxidation Due to the 44.15% and 36.63% DPPH antioxidant activity, decrease after 4 and 20 h respectively, this showed a theoretical uptake of laminarin and fucoidan antioxidant compounds.	[138]
Beef	*Himanthalia elongata*	Washing and cutting Hot air drying Rehydration Seaweed extracts employed 10–40% (*w*/*w*)	Patties with seaweed reduced cooking loss and increased almost 50% tenderness, dietary fibre, total phenolic content, and DPPH radical scavenging activity compared to those without seaweed. Patties with 40% seaweed had highest sensory quality. No bacterial growth and lower lipid oxidation levels detected in patties with ≥20% seaweed	[139]
Chicken	*Himanthalia elongata*	Powdered seaweed employed: 3% dry matter	Seaweed addition decreased the cooking loss. Products with seaweed had higher levels of total viable counts, lactic acid bacteria, tyramine, and spermidine. Seaweed incorporated products have the potential to maintain the desired properties with low salt content.	[140]
Salmon	*Saccharina Latissima*	Storing in flow-through seawater Vacuum packaging Freezing and thawing Salmon wrapped with seaweed biomass and soaked in thawing liquid employed in the ratio of 5:2.5:2.5	Seaweed processed product decreased off-smelling compound with an increase of umami-related compounds. Taste and shelf life were improved with the salmon in conjunction of seaweed.	[141]
Bread	*Fucus vesiculosus*	Tray drying Milling Powdered seaweed employed: 2, 4, 6, and 8% (flour basis)	The addition of seaweed significantly modified wheat dough and bread properties. Over 4% seaweed powder addition had negative effect on dough final porosity and final colour.	[142]
Gluten-free pasta	*Laminaria ochroleuca*	Dehydrated. Milled. Sieved to <0.24 mm and 0.25–2 mm particle size.	The addition of seaweed into gluten-free pasta had similar mechanical and textural characteristics as the control pasta. Mineral and fiber content were increased in the seaweed-based pasta.	[143]
Noodle	*Gracilaria seaweed*	Washing Freeze drying Extracted seaweed powder employed: 0, 1, 3, 5, and 7% (flour basis)	Noodle with 3% seaweed significantly increased their total dietary fiber content. Sensory evaluation showed a moderate acceptability level among consumers.	[144]
Milk	*Ascophyllum nodosum & Fucus vesiculosus*	Washing and freezing Freeze drying and extracting Extracted seaweed was employed: 0.25% and 0.5% (*w*/*w*)	Seaweed extract incorporated milk had potential improving certain milk quality and shelf life. Seaweed extracts were stable in milk and led to various antioxidant activity (DPPH and FICA) before and after in vitro digestion.	[145]
Spice	*Kappaphycus alverezii*	Washing Cross flow drying and grinding Seaweed powder employed with refined vegetable oil in ratios of 15:30, 20:40, and 25:45	Spice with 20% seaweed had high consumer acceptability. Incorporation of seaweed powder to spice improved ash, protein, crude fiber, vitamin E, niacin, vitamin B2 except vitamin B1, and vitamin A.	[146]

## Data Availability

Not applicable.

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
