# Peer review of "Biomolecules from Macroalgae—Nutritional Profile and Bioactives for Novel Food Product Development"

_biomolecules, 2023, doi:10.3390/biom13020386_

Round 1
Reviewer 1 Report (New Reviewer)
The manuscript is well struture and presents a comphrensive revision regarding the aspects of macroalge as food ingredient. Recently, the European Comission released a list of edible algae. I did not see regulation aspects in the present manuscript. As alternative source of protein are a trend topic an not all alage are editble I would like to see some regulation aspects regarding the use of alage in food industry as well as a list of approved species for human nutrion.
Author Response
Dear reviewer,
Thank you very much for your feedback. It is much appreciated. We agree, the legislation on novel protein sources is a very important matter, and very topical. We have now included a paragraph relating to this document in the final section of the manuscript.
Thank you
Laura
Reviewer 2 Report (Previous Reviewer 3)
The review manuscript “Biomolecules from Macroalgae – Nutritional Profile and Bioactives for Novel Food Product Development” (biomolecules-2079674) showed a critical and actual review of seaweed. This manuscript is interesting and discussion showed many beneficially to health to use seaweeds, as I reported last time. However, the author not consiering my suggestion. Basically all the notes I made were not considered. So I will add similar comments again about this manuscript.
The authors have done a good work, with many relevance research’s and perceptively to seaweed, which employing various references based on a scientific method and structure. The manuscript is good and minor points its necessary by adjusting in text. In addition, this manuscript demonstrated great and relevant to reader to Biomolecules. I’ think, the manuscript can be accept to Biomolecules journal after minor revision. Figure quality were good, and acceptable for publication.
Please, use to template “Microsoft or Latex Template” of manuscript to figures, tables and references need adjust following “Author Instructions in Biomolecules”.
-Tables and figures, need correction following “Author Instructions” and displayed after first citation.
- My suggestion to authors. Add um topic about economical seaweed, fo example, coast production in aquatic farms. Maybe, has been interesting to readers by Biomolecules journal.
-I'm suggesting legends to Figure 1. “Scheme” or “Flowchart” of seaweed properties, processes in seaweed aquaculture and products
Minor points:
-Title: only the first capital letter of each word;
-Alphabetic order keywords;
-Please, note 59-120 g/kg-1 dw or 18.4-55.34 mg/g is not correct. Modify by xx-xx g kg-1 dw; yy-yy mg g-1;
-Table 1 - Line in top;
-Legend table 2 - mg/g; mg g-1;
-L59 - (Ramadhani et al.); year?; (ii) seaweed processes
-L96-97 and 100-117. Why italic? Check in all manuscript. I don’t understand.
-L210 kg-1;
-L280. Its not table are box. Please, modify legend or adjust to table;
-L383 and 386, scientific notation;
-L440, year?
-Table 2; bad formatting;
-Please, check table 5, scientific units;
-4.1 topic; formatting;
-Reference and sentence, please check; “or exploitation by food industry (R. Kumar 49 et al. 2021).”.
- In topic 4.5; I suggesting to authors, update references until five years; Example, Alvarenga et al. 2011 to more recent;
-Topic 4.7, Italic?
- Please. All standardization of nomenclature and scientific notation. Check all manuscript.
Best Regards
Author Response
Dear Reviewer,
All your comments have been taken on board and the manuscript updated accordingly. Apologies for the delay in this. We made these changes previously but uploaded the wrong manuscript in error.
Please see the attachment.
Thanks

Round 2
Reviewer 2 Report (Previous Reviewer 3)
I consider the authors made important changes in the manuscript and it was highly improved. I recommend the publication of the manuscript in its current form. Best Regards
This manuscript is a resubmission of an earlier submission. The following is a list of the peer review reports and author responses from that submission.
Round 1
Reviewer 1 Report
The study is a review of the nutritional potential of seaweed. While the topic is interesting, the text is very difficult to understand. Sentences without much context are linked together, making it difficult to understand. The paragraphs are very long and cover different topics.
Tables 4 and 5 are exactly the same as in another paper. This can be considered plagiarism (in this case, self-plagiarism).
Also, this is a review paper and it is not clear how the information was extracted. Therefore, it is not possible to judge the quality of the information.
There are many sentence without references.
In which extent this paper is different from the others published by the group (and other groups)?
Comments:
Do not you think "seaweed" should be included in the title?
L27 - 31 - This sentence definitely needs references.
The first paragraph of the introduction is very messy. It mixes nutritional value with bioactive properties and consumption. I suggest the authors to better divide this paragraph into three sections (at least):
1. what is seaweed?
2. What nutritional properties have been researched? How is it consumed?
3.Why is this review relevant - How was it constructed?
L59 I think there is a (ii) missing.
L59 What is the citation here? (Ramadhani et al.)
What type of review is it? No method or procedure is described in the paper. Is it a narrative review? Please be more clear.
A table in the introduction is a bit odd... I think you could move table 1 and 2 to section 2.1.1.
The whole text is very hard to read. The paragraphs are very long and connect different sentences.
L268 - This sentence needs references
Table 3 is not formatted correctly
Tables 4 and 5 were not taken from other studies, but are exactly the same tables from the other studies. This can be considered as self-plagiarism.
Reviewer 2 Report
This review article is presenting the properties of the seaweed as foods, the characteristics of the processes involved in the seaweed aquaculture as well as of the main derived products. The review is well structured, logical and presents in a manner that arouses the reader's interest some of the challenges related with this type of seaweed as food (the consumers hesitation, food safety issues or as some processing hurdles). The article can be accepted in the present form.
Reviewer 3 Report
The review manuscript “Biomolecules from Macroalgae – Nutritional Profile and Bioactives for Novel Food Product Development” (biomolecules-1974577) showed a critical and actual review of seaweed. This manuscript is interesting and discussion showed many beneficially to health to use seaweeds.
The authors have done a good work, with many relevance research’s and perceptively to seaweed, which employing various references based on a scientific method and structure. The manuscript is good and minor points its necessary by adjusting in text. In addition, this manuscript demonstrated great and relevant to reader to Biomolecules. I’ think, the manuscript can be accept to Biomolecules journal after minor revision. Figure quality were good, and acceptable for publication.
Please, use to template “Microsoft or Latex Template” of manuscript to figures, tables and references need adjust following “Author Instructions in Biomolecules”.
-Tables and figures, need correction following “Author Instructions” and displayed after first citation.
- My suggestion to authors. Add um topic about economical seaweed, fo example, coast production in aquatic farms. Maybe, has been interesting to readers by Biomolecules journal.
-I'm suggesting legends to Figure 1. “Scheme” or “Flowchart” of seaweed properties, processes in seaweed aquaculture and products
Minor points:
-Title: only the first capital letter of each word;
-Alphabetic order keywords;
-Please, note 59-120 g/kg-1 dw or 18.4-55.34 mg/g is not correct. Modify by xx-xx g kg-1 dw; yy-yy mg g-1;
-Table 1 - Line in top;
-Legend table 2 - mg/g; mg g-1;
-L59 - (Ramadhani et al.); year?; (ii) seaweed processes
-L96-97 and 100-117. Why italic? Check in all manuscript. I don’t understand.
-L210 kg-1;
-L280. Its not table are box. Please, modify legend or adjust to table;
-L383 and 386, scientific notation;
-L440, year?
-Table 2; bad formatting;
-Please, check table 5, scientific units;
-4.1 topic; formatting;
-Reference and sentence, please check; “or exploitation by food industry (R. Kumar 49 et al. 2021).”.
- In topic 4.5; I suggesting to authors, update references until five years; Example, Alvarenga et al. 2011 to more recent;
-Topic 4.7, Italic?
- Please. All standardization of nomenclature and scientific notation. Check all manuscript.
Best Regards